# Long-Acting Recombinant IL-7 (rhIL-7-hyFc) Enhances the Primary and Memory Neoantigen-Specific Immune Response to Breast Cancer Personalized Cancer Vaccines

**DOI:** 10.3390/cancers17193177

**Published:** 2025-09-30

**Authors:** Michael Chen, Thomas Kane, Ina Chen, Suangson Supabphol, Xiuli Zhang, Alexandra A. Wolfarth, Donghoon Choi, Lijin Li, S. Peter Goedegebuure, William E. Gillanders

**Affiliations:** 1Department of Surgery, Washington University School of Medicine, Saint Louis, MO 63110, USAssupabphol@hotmail.com (S.S.); xiulizhang@wustl.edu (X.Z.);; 2NeoImmuneTech, Inc., Rockville, MD 20850, USA; 3The Alvin J. Siteman Cancer Center, Washington University School of Medicine and Barnes-Jewish Hospital, Saint Louis, MO 63110, USA

**Keywords:** DNA vaccine, immune response, neoantigen, recombinant IL-7, breast cancer

## Abstract

**Simple Summary:**

Personalized cancer vaccines (PCVs) can train the immune system to attack tumors by targeting neoantigens, but their effectiveness depends on the strength and durability of the immune response. In this study, we tested whether a long-acting form of interleukin-7 (rhIL-7-hyFc), a molecule known to support T cell survival, could improve the performance of a DNA-based PCV in a mouse breast cancer model. We found that combining rhIL-7-hyFc with the vaccine led to stronger and longer-lasting CD8+ T cell responses, better immune memory, and more tumor-infiltrating lymphocytes. This combination also helped prevent tumor growth more effectively. Our results suggest that rhIL-7-hyFc could be a useful addition to PCV-based cancer treatments.

**Abstract:**

**Background:** Personalized cancer vaccines (PCVs) are a promising form of cancer immunotherapy, capable of eliciting robust neoantigen-specific immune responses. However, cancer neoantigens are variable in terms of immunogenicity, and PCVs may be less effective when targeting weak neoantigens. Strong and durable immune responses are also likely to be critical for vaccine efficacy. Interleukin-7 (IL-7) is a common gamma-chain cytokine known to support T cell development and survival, and a long-acting form of recombinant human IL-7 fused with hybrid Fc (rhIL-7-hyFc) has shown potential to enhance immune responses in early-stage clinical trials. **Methods:** In this study, we evaluated the ability of rhIL-7-hyFc to serve as a molecular adjuvant to a DNA PCV in the E0771 murine breast cancer model. **Results**: We found that the combination of rhIL-7-hyFc and DNA PCV treatment prolonged neoantigen-specific CD8+ T cell responses, improved functional memory as measured based on in vivo cytotoxicity, and increased the number of neoantigen-specific tumor-infiltrating lymphocytes (TILs), resulting in improved prophylactic tumor protection and durable memory responses. **Conclusions:** Our findings support the potential of rhIL-7-hyFc to enhance the efficacy of PCVs and suggest clinical utility for adjuvant rhIL-7-hyFc in cancer immunotherapy.

## 1. Introduction

Cancer neoantigens are tumor-specific antigens considered to be promising targets for cancer immunotherapy. Neoantigens typically arise from genetic alterations in protein-coding regions, but they can also originate from disordered post-transcriptional and post-translational modifications, aberrant RNA splicing, or proteasomal misprocessing [1,2,3,4,5]. Unlike tumor-associated antigens (e.g., PSA, HER2, MUC1), neoantigens are not expressed in normal tissues and are therefore not subject to central mechanisms of immune tolerance, making them highly attractive targets for immunotherapy [5,6]. Various bioinformatic approaches exist to identify candidate neoantigens for personalized vaccine therapy [3,7,8,9]. Research from our group and others has validated these approaches both in pre-clinical and clinical settings [4,5,7,10]. Personalized cancer vaccines (PCVs) possess the ability to expand T cell populations toward specific target antigens, have proven to be safe in clinical translation, and in initial studies, are associated with prolonged survival in cancer patient populations [1,7,10].

Various platforms for PCV delivery have been explored (i.e., peptide, DNA, and mRNA) [1,4,10,11], and efforts are ongoing to identify the most effective platform. While capable of eliciting robust immune responses, there is a concern that PCVs may be associated with less robust long-term CD4+ and CD8+ T cell immunity, which may impair long term efficacy [12]. Generally, PCVs target multiple tumor neoantigens in a single vaccine. However, typically only a subset of the included neoantigens elicit immune recognition [10,12,13]. The remaining, weaker neoantigens contribute little to tumor-specific immunity, thereby reducing the potential for tumor protection [10]. Despite these challenges, PCVs have shown significant promise in clinical studies. Emerging research suggests that combining PCVs with other immunotherapies may enhance their efficacy [7,11,14]; however, no current combination therapy simultaneously addresses both limitations—namely the ability to enhance the response to weak neoantigens and prolong the response.

Interleukin-7 (IL-7) is a common gamma-chain cytokine that has been investigated to improve and sustain vaccine-mediated immunity [15]. Biologically, IL-7 plays a key role in the development and differentiation of naïve T cells, while also supporting memory T cell survival [16,17]. By activating the STAT3 and STAT5 signaling pathways, IL-7 supports effector memory CD8+ T cell persistence [18,19]. IL-7 also demonstrates a capacity to increase the overall T cell repertoire, amplifying immune diversity and arming the adaptive immune system to face a broader range of antigens [16,20]. While mechanistically IL-7 promotes several functions to enhance adaptive immunity, its short half-life in vivo [16,21] has greatly limited its utility as a pharmaceutical agent, encouraging investigation into more durable IL-7 variants [21]. 

Fc-fused long-acting recombinant human IL-7 (rhIL-7-hyFc, efineptakin alfa, or NT-I7) was engineered to extend IL-7′s half-life and improve its therapeutic potential [21]. rhIL-7-hyFc differs from endogenous IL-7 in that it is fused to an antibody fragment crystallizable (Fc) region, stabilizing the cytokine in vivo and reducing the chance of complement activation [22]. Much like IL-7, rhIL-7-hyFc has been shown to enhance CD8+ effector and memory T cells without increasing T regs in the tumor microenvironment (TME), optimally priming the immune system [21,23]. Validating this platform in mice, Kim et al. [22] demonstrated that an rhIL-7-hyFc adjuvant significantly improved CAR-T therapy by enhancing memory T cells, increasing CD4+ effector T cell prevalence, and reducing T cell exhaustion. Currently, several phase I and phase II clinical trials are underway investigating adjuvant rhIL-7-hyFc (NT-I7) to treat cancers, alongside radiotherapy and additional immunotherapies (i.e., temozolomide, CAR T therapy, checkpoint inhibitors) (NCT04781309, NCT07052305, NCT04332653, NCT04984811, NCT05075603, NCT05600920, NCT05465954) [22]. Early findings from completed Phase I studies suggest rhIL-7-hyFc offers a strong safety and efficacy profile, furthering the adjuvant’s translational excitement [24]. To date, rhIL-7-hyFc has not been investigated in combination with PCVs. In this study, we aim to improve the efficacy of PCVs by investigating rhIL-7-hyFc in combination with a validated polyepitope DNA PCV in the E0771 murine model.

## 2. Materials and Methods

### 2.1. Animals

Female C57BL/6 mice (stock number 000664) were purchased from The Jackson Laboratory (Bar Harbor, ME, USA). All experiments were performed on animals 6–10 weeks of age. Animal studies were approved by the Institutional Animal Studies Committee of Washington University School of Medicine. Masking or blinding was not implemented in this study.

### 2.2. Tumor Cell Line

E0771 is a murine breast cancer cell line of C57BL/6 (H-2b) origin [25]. The cell line was carried in RPM1640 media (Gibco, Billings, MT, USA) supplemented with 10% FBS (Atlanta Biologicals, Oakwood, GA, USA), 1% penicillin-streptomycin, L-glutamine, sodium pyruvate, and non-essential amino acid (Gibco, Billings, MT, USA).

### 2.3. Polyepitope Neoantigen DNA Vaccination

Neoantigens were identified and cloned into the mammalian expression plasmid pcDNA 3.1(+) as previously described [26]. Polyepitope plasmid DNA was amplified in Escherichia coli DH5α (Invitrogen, Waltham, MA, USA) and purified using the GenElute HP Plasmid Maxiprep Kit (Millipore Sigma, Burlington, MA, USA). PCVs were performed using a Helios gene gun (Bio-Rad, Hercules, CA, USA) as previously described [26]. Mice were vaccinated three times, three days apart, with 4 µg of DNA delivered per dose [27].

### 2.4. rhIL-7-hyFc Treatment

rhIL-7-hyFc (5 mg/kg) was administered subcutaneously on day 4 or day 13 post-PCV, during the T cell expansion or contraction phase, respectively.

### 2.5. ELISpot

Murine spleens were manually dissociated and filtered through a 70 µm filter. RBC lysis was performed with ACK lysing buffer (ThermoFisher, Waltham, MA, USA). IFN-γ ELISpotPLUS Kits (Mabtech, Cincinnati, OH, USA) were used as per the manufacturer’s instructions; 2.5 × 10^5^ splenocytes were plated per well in triplicate and stimulated with neoantigen peptides (GenScript, Piscataway, NJ, USA) for 18 hours. ELISpot plates were scanned and analyzed on an ImmunoSpot Reader (CTL, Shanker Heights, OH, USA). 

### 2.6. Tumor Challenge

E0771 tumor cells were dissociated using 0.05% Trypsin/EDTA (ThermoFisher, Waltham, MA, USA) and washed twice with Ca^2+^/Mg^2+^-free PBS. Then, 10^5^ cells were injected subcutaneously into the flank of each mouse. The tumor length and width were measured approximately every 3 days with an electronic caliper. Tumor volumes were calculated with the formula volume = ½ (L × W^2^).

### 2.7. In Vivo Cytotoxicity Assay

In vivo cytotoxicity assays were performed as previously described, and CFSE concentrations were well within the typical range reported in the literature [28,29]. Briefly, syngeneic naïve splenocytes were pulsed with or without neoantigen peptides and labeled with 0.5 µM (CFSE^lo^) or 5 µM (CFSE^hi^) CFSE (ThermoFisher, Waltham, MA, USA), respectively. Labeled splenocytes were subsequently injected intravenously into vaccinated mice. Sixteen hours post-injection, mice were sacrificed, and the spleen was harvested for flow cytometry analysis (BD FACScalibur, Milpitas, CA, USA). Neoantigen-specific cytotoxicity was calculated as follows: % specific lysis = (1 − %CFSE^lo^/%CFSE^hi^) × 100.

### 2.8. Flow Cytometry TIL Analysis

Tumors were removed from mice and dissociated with a tumor dissociation kit (Miltenyi Biotec, Bergisch Gladbach, Germany). CD4+ and CD8+ TILs were isolated using CD4/CD8 Microbeads (Miltenyi Biotec, Bergisch Gladbach, Germany). Isolated TILs were Fc receptor blocked using the TruStain FcX antibody (Biolegend, San Diego, CA, USA) and labeled with APC-conjugated Lrrc27/H-2Db dextramer (Immudex, Copenhagen, Denmark) and subsequently surface stained for CD3e and CD8ɑ surface markers. Data was acquired using a BD Fortessa X-20 (BD, Milpitas, CA, USA) and analyzed using Flowjo v10 software. 

### 2.9. Statistics

Data were analyzed using GraphPad Prism software version 10 and presented mainly as the mean ± SEM. All data generated in the animal studies were included in the analysis. For groups that were not assumed to have equal variances, Mann–Whitney or one-way ANOVA was applied. A *p*-value equal to or less than 0.05 was considered statistically significant.

## 3. Results

### 3.1. rhIL-7-hyFc Administered During the T Cell Expansion or Contraction Phase Prolongs Neoantigen-Specific T Cell Immunity

To define the kinetics of the immune response following a PCV DNA vaccine, C57BL/6 mice were vaccinated with neoantigen PCV (4 µg DNA delivered via a Helios gene gun on days 0, 3, and 6). ELISpot assays were performed every 3rd day, starting 8 days after the start of vaccination (Appendix A). Following vaccination, neoantigen-specific T cells increased, resulting in a peak at day 11 followed by a decrease back to baseline by day 17 (Appendix A). This pattern suggested that T cells expand between the start of immunization to day 11 (T cell expansion phase), and at the time between day 11 and day 17, T cells were in the contraction phase.

We subsequently administered rhIL-7-hyFc during the T cell expansion phase (day 4) or T cell contraction phase (day 13) to determine the optimal timing of administration. rhIL-7-hyFc (5 mg/kg) was administered subcutaneously on day 4 or day 13 after the start of the PCV (Figure 1a,c). rhIL-7-hyFc at 5 or 10 mg/kg is routinely used in mouse preclinical models with no known toxicity issues [30,31]. Additionally, rhIL-7-hyFc is well tolerated in patients in the range of 60–1200 µg/kg [32]. The dose level of 5 mg/kg rhIL-7-hyFc in mice is equivalent to 405 µg/kg in humans, which is a clinically relevant dose. ELISpot assays were performed on days 11 and 20 to measure neoantigen-specific T cell immunity. The administration of rhIL-7-hyFc on day 4 did not significantly enhance neoantigen-specific responses on day 11 compared to the PCV alone (Figure 1b). The addition of rhIL-7-hyFc to DNA PCV during the expansion or contraction phase enhanced T cell responses to the neoantigen Lrrc27 equally at day 20. Interestingly, only the administration of rhIL-7-hyFc during the contraction phase significantly increased responses to the neoantigens Pttg1 and Plekho1 at day 20 (Figure 1d). Thus, we determined the administration of rhIL-7-hyFc during the contraction phase to be optimal for enhancing durable PCV-induced neoantigen responses, even among relatively weak neoantigens: Pttg1 and Plekho1.

### 3.2. PCV + rhIL-7-hyFc Protects from Tumor Challenge

C57BL/6 mice were vaccinated with the neoantigen PCV on days 0, 3, and 6 with and without rhIL-7-hyFc according to the optimized regimen (Figure 2a). Twenty-seven days after the start of vaccination, mice were challenged with 10^5^ E0771 tumor cells subcutaneously. The PCV and rhIL-7-hyFc combination treatment conferred the greatest tumor protection, while PCV alone offered no protection compared to the control (Figure 2b). Interestingly, rhIL-7-hyFc alone conferred modest tumor protection suggesting an enhanced non-vaccine specific immunologic response to the tumor. 

### 3.3. PCV + rhIL-7-hyFc Generates Stronger Neoantigen-Specific Killing and Memory CD8+ T Cell Responses

C57BL/6 mice were vaccinated with the neoantigen PCV on days 0, 3, and 6. rhIL-7-hyFc (5 mg/kg) was administered on day 13 as previously optimized (Figure 3a). Neoantigen-specific killing was measured through an in vivo killing assay using naïve splenocytes labeled at two CFSE concentrations. Cells labeled with the lower concentration of CFSE were pulsed with the minimal MHC class I-restricted peptides corresponding to the PTTG1, PLEKHO1, and Lrrc1 neoantigens. Cells labeled with the higher concentration of CFSE were not pulsed with peptide. At day 22, mice that received the PCV and rhIL-7-hyFc demonstrated 45% killing, compared to 7% killing in mice that received the PCV alone. The neoantigen-specific killing induced by the PCV in combination with rhIL-7-hyFc was durable, with 23% killing observed at day 51 (Figure 3b,c).

### 3.4. Combination of PCV and rhIL-7-hyFc Increases Neoantigen-Specific TILs

The enhancement of neoantigen-specific T cell immunity by rhIL-7-hyFc after vaccination was further confirmed with Lrrc27 dextramer staining of TILs. Mice were challenged with 10^5^ E0771 tumor cells. Six days after tumor inoculation, when tumor sizes were approximately 50 mm^3^, mice were treated with the neoantigen PCV with or without rhIL-7-hyFc (Figure 4a). Tumor growth kinetics for this model required modification of the treatment schedule, with rhIL-7-hyFc treatment given during the T cell expansion phase (day 4). Tumors were harvested on day 11 after the start of treatment, and TILs were isolated. Flow cytometry analysis was performed to identify Lrrc27-neoantigen-specific CD8+ T cells using the Lrrc27 dextramer. Lrrc27 was used for dextramer staining since it is the most immunogenic neoantigen based on ELISpot data. The PCV of tumor-bearing mice was found to result in 2.5% CD8+ TILs specific for Lrrc27. The portion of Lrrc27 CD8+ TILs increased to 12% when the PCV was given in combination with rhIL-7-hyFc (Figure 4b,c).

## 4. Discussion

In this study, we report that the combination of PCV + rhIL-7-hyFc treatment significantly prolonged the immune recognition of sub-dominant neoantigens, promoted sustained neoantigen-specific cytotoxic activity, and prevented tumor growth in vivo. We also observed that combination treatment increased neoantigen-specific tumor infiltrating lymphocytes (TILs). When combined with PCV, rhIL-7-hyFc exhibited a strong adjuvant profile, with combination rhIL-7-hyFc + PCV treatment enhancing anti-tumor activity via a more robust and durable neoantigen-specific T cell immunity. Our findings confirm rhIL-7-hyFc’s utility as an immunotherapy adjuvant and support a novel approach for improving PCV efficacy.

Common gamma-chain cytokines—including IL-2, IL-7, IL-15, and IL-21—are key regulators of T cell homeostasis and function and have been widely explored as immunotherapeutic agents [21]. As of 2022, over 150 clinical trials have investigated these cytokines as monotherapies, in combination with checkpoint inhibitors, as potential adjuvants to CAR-T therapies, and in combination with cancer vaccines [21]. Among gamma-chain cytokines, IL-7 stands out for its broad potential to support T lymphocyte development and proliferation without triggering adverse immunoregulatory effects [17,20,33]. Mechanistically, IL-7 activates JAK family kinases, which stimulate STAT transcription factors, driving gene expression that promotes downstream CD4+ and CD8+ T cell expansion [17,20,33]. After T cell expansion, IL-7 has been shown to support long-term T cell survival by modulating apoptotic and metabolic pathways [34]. Clinically, native IL-7 has been met with inconsistent success when investigated as an enhancer of oncologic and immunologic interventions [18,24,35], largely due to its rapid in vivo clearance [21]. With the recent development of recombinant human IL-7 fused to a hybrid Fc domain (rhIL-7-hyFc, NT-I7), the therapeutic potential of IL-7 has increased significantly. Importantly, rhIL-7-hyFc significantly prolongs IL-7′s half-life, without compromising IL-7′s vital regulatory functions [23,33]. rhIL-7-hyFc has since been investigated in clinical studies as a monotherapy and immunotherapy adjuvant, with its safety and activity being confirmed in multiple Phase I trials [24,36,37].

Previously published CAR-T literature first established a role for rhIL-7-hyFc in enhancing T cell responses in the context of suboptimal TCR–antigen interactions by promoting a larger and more diverse T cell pool [22,38]. Related studies have shown rhIL-7-hyFc can also expand naïve T cells and increase TCR diversity [36,37,38]. Taken together, strong, literature-supported rationale suggests that when used in conjunction with PCVs, rhIL-7-hyFc can increase the immune recognition of weaker neoantigens and lower the threshold for weak-neoantigen-specific T cell proliferation. In mice administered PCV and rhIL-7-hyFc combination treatment, we observed a significant improvement in long-term neoantigen-specific killing and memory CD8+ T cell formation arising from two relatively weak neoantigens (Figure 1 and Figure 3). These mechanistic findings translated to a substantial improvement in rhIL-7-hyFc + PCV-mediated tumor protection compared to the PCV alone (Figure 2). Our results support rhIL-7-hyFc’s ability to enhance relatively weak antigen immune recognition and suggest a novel strategy for targeting tumor-specific antigens. With rhIL-7-hyFc’s safety and activity profile previously established in clinical settings, our findings suggest the immunomodulator as an attractive candidate for addressing two known PCV clinical shortcomings: (1) rhIL-7-hyFc stimulated long-term PCV-induced neoantigen immune recognition, (2) while increasing the repertoire of durably immunogenic neoantigens to include two relatively weaker antigens.

The rationale for administering adjuvant rhIL-7-hyFc in lymphopenic cancer populations is especially strong. Lymphopenia—a reduction in circulating lymphocytes—is a negative prognostic marker for several cancers, including triple-negative breast cancer (TNBC) and glioblastoma (GBM) [36,39,40,41]. In a trial of 18 patients with GBM, rhIL-7-hyFc was found to safely and effectively restore lymphocyte counts without inducing adverse immune regulatory effects [36]. In our preclinical model, we similarly observed lymphoid expansion, evidenced by an increase in spleen size among combination rhIL-7-hyFc PCV-treated groups (Appendix A), which is a well-known rhIL-7-hyFc-induced effect [23,36]. Lymphopenia in cancer populations poses a significant challenge for PCV therapy, due to the PCV’s reliance on intact adaptive immunity. Our findings support a unique clinical application for combining rhIL-7-hyFc with existing PCV therapies, particularly for cancers known to induce lymphopenia (i.e., GBM, TNBC), due to the cytokine’s demonstrated ability to safely rescue total lymphocyte counts, broaden the T cell repertoire, and prolong neoantigen-specific memory T cells. Thus, rhIL-7-hyFc has the potential to not only broaden the repertoire of immunogenic neoantigens but also broaden the pool of promising patient candidates for PCVs. 

A limitation of this study is that rhIL-7-hyFc was administered only once, preventing an evaluation of its immunologic effects following repeated dosing. In clinical trials, rhIL-7-hyFc is typically administered at multiple timepoints during a treatment cycle, often at 6–12 week intervals [24,42,43]. However, the repeated administration of high-dose recombinant human cytokines in mice has been shown to induce neutralizing effects [44]. To minimize the risk of autoimmunity, we limited rhIL-7-hyFc administration to a single dose in our models. Consequently, by studying rhIL-7-hyFc in C57BL/6 mice under these conditions, we may not capture the molecule’s full immune potential. Also worth noting, our study evaluated rhIL-7-hyFc exclusively in combination with a DNA-based PCV. The immunogenic neoantigens in the E0771 model all elicited CD8 T cell responses. Although rhIL-7-hyFc is also known to enhance CD4 T cell responses, we were not able to test this in the E0771 model system. We provide strong evidence that rhIL-7-hyFc can enhance PCV efficacy; however, with vaccine delivery technology rapidly advancing, investigating rhIL-7-hyFc with other neoantigen delivery platforms—such as mRNA—could further optimize PCV development. Our findings provide proof-of-concept that rhIL-7-hyFc enhances the impact of PCV. However, our data was obtained using a single mouse breast cancer model and may not be generalizable to other models or to patients. Lastly, an important area for future study will include a broader investigation of other immune subsets to determine whether rhIL-7-hyFc modulates PCV-stimulated immune populations beyond CD8 T cells.

## 5. Conclusions

Here, we present long-acting rh-IL-7-hyFc (NT-I7) as an adjuvant to a validated polyepitope neoantigen PCV. Our preclinical findings suggest that rhIL-7-hyFc + DNA PCV treatment increases the duration of neoantigen-specific anti-tumor immunity, broadens the repertoire of immunogenic neoantigens, protects against tumor challenge prophylactically, and promotes neoantigen specific TILs therapeutically. Our study encourages future research investigating adjuvant rhIL-7-hyFc in clinical settings.

## Figures and Tables

**Figure 1 cancers-17-03177-f001:**
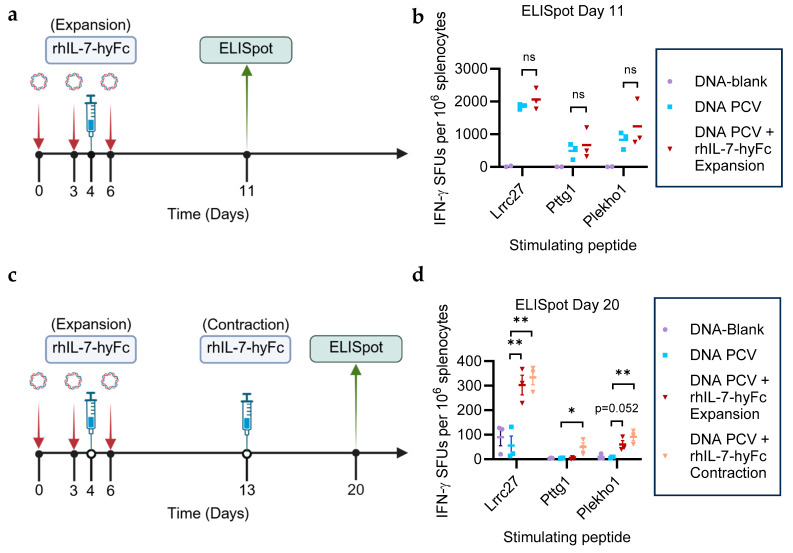
rhIL-7-hyFc prolongs neoantigen-specific T cell immunity. (**a**) Mice were vaccinated with the DNA PCV as per the protocol on days 0, 3, and 6. rhIL-7-hyFc was given during the T cell expansion phase on day 4. An ELISpot assay was performed on day 11. (**b**) Administering rhIL-7-hyFc on day 4 in addition to PCV shows a comparable T cell immune response compared to PCV alone on day 11 (n = 3 each group). (**c**) Mice were vaccinated with DNA PCV as per the protocol on days 0, 3, and 6. rhIL-7-hyFc was given during the T cell expansion or contraction phase on days 4 or 13, respectively. An ELISpot assay was performed on day 20. (**d**) rhIL-7-hyFc given on days 4 or day 13 in combination with PCV increased neoantigen-specific T cell responses measured via ELISpot on day 20 compared to PCV alone (n = 3 each group). *: *p* ≤ 0.05; **: *p* ≤ 0.01.

**Figure 2 cancers-17-03177-f002:**
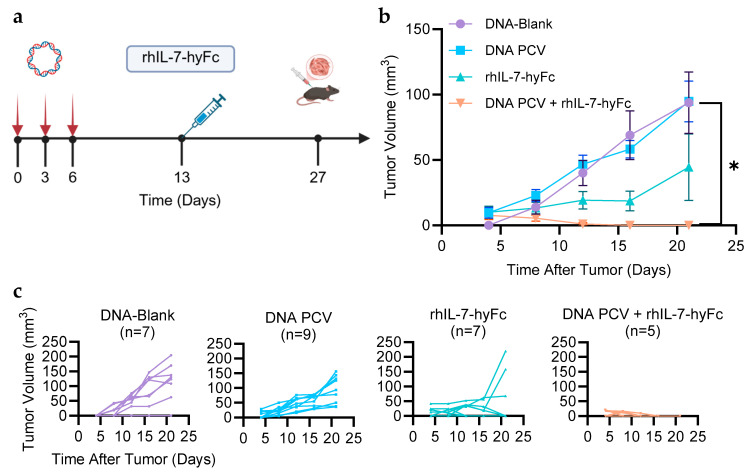
PCV in combination with rhIL-7-hyFc protects against tumor growth. (**a**) Mice were treated with the empty vector plasmid DNA or DNA PCV on days 0, 3, and 6, and 5 mg/kg rhIL-7-hyFc was administered subcutaneously on day 13. On day 27, 10^5^ E0771 cells were inoculated subcutaneously into each mouse’s flank. The length (l) and width (w) of the tumor was measured every 4 days after tumor inoculation. The tumor volume for each group was calculated using the formula v = (l × w^2^)/2 and graphed as the average (**b**) or individually (**c**) for each group. The PCV in combination with rhIL-7-hyFc significantly reduced tumor growth compared to PCV alone (*p* = 0.0013). *: *p* ≤ 0.05.

**Figure 3 cancers-17-03177-f003:**
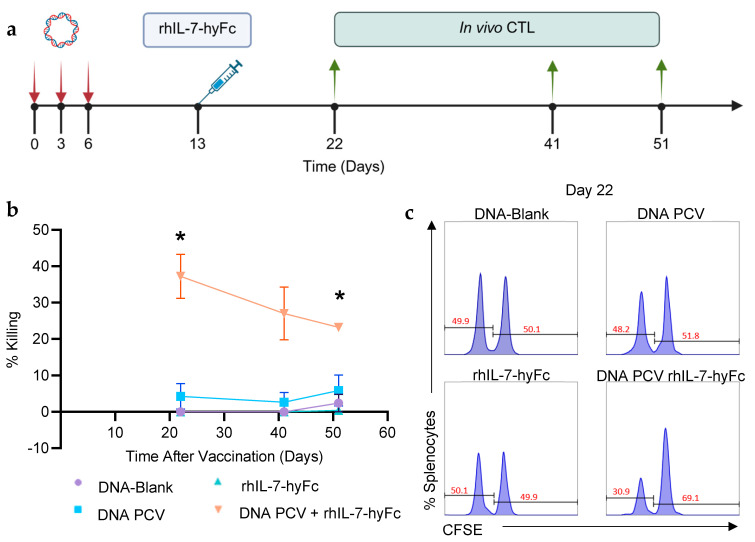
PCV + rhIL-7-hyFc generates stronger neoantigen-specific killing and memory CD8+ T cell responses. (**a**) The PCV with or without rhIL-7-hyFc was administered as per previously optimized protocols. In vivo cytotoxicity assays were performed on days 22, 41, and 51. (**b**) Combination of PCV + rhIL-7-hyFc produced 37% neoantigen-specific killing of pulsed splenocytes, as measured via an in vivo cytotoxicity assay, compared to only 12% by the PCV alone on day 22. Neoantigen-specific killing is long-lasting, with 23% killing even at 51 days after the start of vaccination. A total of 27 animals were used in this experiment (n = 3 per group at each time point). rhIL-7-hyFc group data points nearly mirror the *x*-axis. (**c**) Representative flow cytometry diagrams on day 22. *: *p* ≤ 0.05.

**Figure 4 cancers-17-03177-f004:**
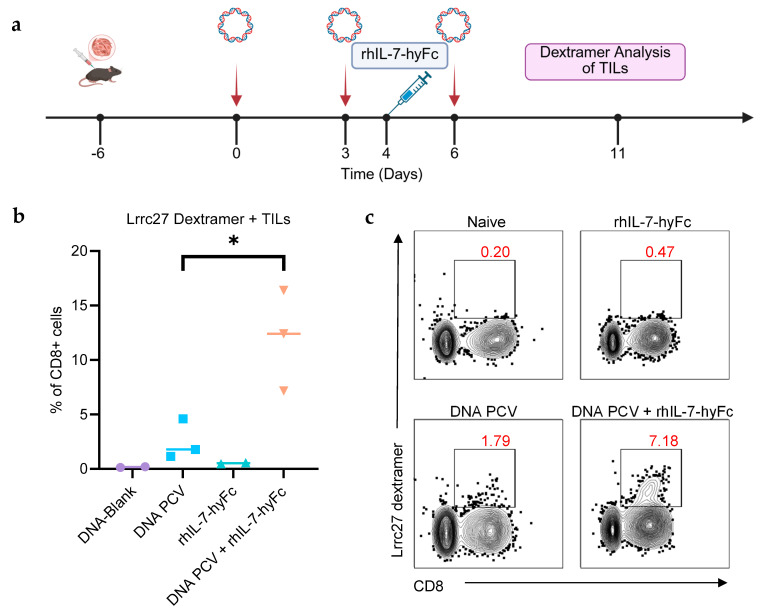
Combination of PCV and rhIL-7-hyFc increases neoantigen-specific TILs. (**a**) Here, 10^5^ E0771 cells were inoculated subcutaneously into the flanks of mice and allowed to grow for 6 days, reaching a volume of approximately 50 mm^3^. Tumor-bearing mice (n = 10 total) were arbitrarily assigned to four treatment groups: DNA-Blank control (n = 2), DNA-PCV only (n = 3), rhIL-7-hyFc only (n = 2), and DNA PCV + rhIL-7-hyFc (n = 3). The PCV was administered on days 0, 3, and 6. rhIL-7-hyFc was given on day 4. Lrrc27 dextramer analysis of TILs was performed 11 days after the start of vaccination. (**b**) Combination of the PCV and rhIL-7-hyFc induced 12% of CD8+ TILs to be Lrrc27-dextramer-positive. The PCV alone only induced 2.5% of CD8+ TILs to be dextramer-positive. (**c**) Representative flow cytometry plots. *: *p* ≤ 0.05.

## Data Availability

The original contributions presented in this study are included in the article/Appendix A. Further inquiries can be directed to the corresponding author(s).

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
