# Peer review of "Long-Acting Recombinant IL-7 (rhIL-7-hyFc) Enhances the Primary and Memory Neoantigen-Specific Immune Response to Breast Cancer Personalized Cancer Vaccines"

_cancers, 2025, doi:10.3390/cancers17193177_

Round 1
Reviewer 1 Report
Comments and Suggestions for Authors
- Since the study focuses on neoantigen-specific vaccine development specifically in breast cancer, I suggest that the authors revise the title to reflect this focus clearly. The current title may be misleading, as all experiments in the manuscript are conducted using breast cancer cell lines.
- I am curious about the rationale for selecting the concentration of rhIL-7-hyFc (5 mg/kg). Did the authors conduct any titration or dose–response experiments to determine this dosage? In addition, were toxicity and drug tolerance assessments performed in the study? If the selected dose is based on clinical trial data, how does this translate to small animal models, and were corresponding toxicity and tolerance evaluations carried out to confirm its suitability?
- I am wondering how the authors categorized PTTG1 and PLEKHO1 as weak neoantigens. Additionally, based on available literature, PALB2, PTPRS, and ZDHHC16, TP53 have shown potential in eliciting immune responses in breast cancer. I am curious whether the authors observed any relevant expression of these antigens following rhIL-7-hyFc treatment, either alone or in combination with DNA-PCV.
- I appreciate the authors’ cell-killing experiment using CFSE. However, I am still curious about the underlying mechanism does the cell killing occur through a direct effect, or are immune cells becoming cytotoxic and responding more rapidly to eliminate cancer cells? Additionally, it is not clear which specific cell populations were targeted by the treatments, either with rhIL-7-hyFc alone or in combination with DNA-PCV.
- I am unclear about the rationale for using lower concentration CFSE-labeled cells pulsed with E0771, while the higher concentration was not. I suggest the authors clarify this point and provide supporting data to help readers better understand the purpose of this approach.
- Although I am partially convinced by the data showing an increase in Lrrc27 neoantigen-specific CD8⁺ T cells with PCV alone (2.5%), and a further five-fold increase when combined with rhIL-7-hyFc, I would like to see whether the authors also evaluated rhIL-7-hyFc alone. In addition, it would be valuable if the authors could provide this dataset in a vaccination setting (Section 3.3).
- In section 3.4, the authors show an increase in Lrrc27 neoantigen-specific CD8⁺ T cells. I am curious whether the authors also examined other immune cell types, such as macrophages or regulatory T cells, in this context. Could the authors provide information on the status of other Lrrc27-specific immune populations? Additionally, given the promising increase in Lrrc27-specific CD8⁺ T cells, it would be helpful to know whether these cells exhibit enhanced cytotoxic activity compared to non-specific CD8⁺ T cells.
- Some of the figures appear blurry and unclear; improving their clarity would enhance data presentation. Additionally, the Discussion section is relatively weak and would benefit from a more thorough justification of the conclusions based on the presented evidence. Finally, there are some grammatical errors and redundancies throughout the manuscript that should be carefully revised.
- Since this manuscript focuses on neoantigen-specific vaccine development, particularly in breast cancer, I suggest that the authors cite the following relevant article, which specifically addresses neoantigens and vaccine strategies in triple-negative breast cancer (TNBC):
Mukherjee, S., & Chatterjee, K. (2024). DNA damage response and neoantigens: A favorable target for triple-negative breast cancer immunotherapy and vaccine development. In International Review of Cell and Molecular Biology (Vol. 389, pp. 104–152). Academic Press. https://doi.org/10.1016/bs.ircmb.2024.05.001. This reference would strengthen the context and rationale for the current study.
Comments on the Quality of English Language
In a few instances, the spaces between adjacent words are missing. There are also grammatical mistakes. Please check the spelling and grammar. I would suggest that the whole manuscript be thoroughly revised to improve clarity and readability.
Author Response
Reviewer #1
Comment #1: Since the study focuses on neoantigen-specific vaccine development specifically in breast cancer, I suggest that the authors revise the title to reflect this focus clearly. The current title may be misleading, as all experiments in the manuscript are conducted using breast cancer cell lines.
Reply: We have revised the manuscript title to clarify that the manuscript is focused on personalized cancer vaccines in breast cancer. The new title is: "Long-acting recombinant IL-7 (rhIL-7-hyFc) enhances the primary and memory neoantigen-specific immune response to breast cancer personalized cancer vaccines."
Comment #2: I am curious about the rationale for selecting the concentration of rhIL-7-hyFc (5 mg/kg). Did the authors conduct any titration or dose-response experiments to determine this dosage? In addition, were toxicity and drug tolerance assessments performed in the study? If the selected dose is based on clinical trial data, how does this translate to small animal models, and were corresponding toxicity and tolerance evaluations carried out to confirm its suitability?
Reply: The optimal dose of rhIL-7-hyFc in preclinical mouse studies was determined in previous studies. We have revised the manuscript to clarify this issue. Please see the Results Section 3.1: "rhIL-7-hyFc at 5 or 10mg/kg is routinely used in mouse preclinical models with no known toxicity issues [30,31]. Additionally, rhIL-7-hyFc is well tolerated in patients in the range of 60-1200ug/kg [32]. The dose level of 5mg/kg rhIL-7-hyFc in mice is equivalent to 405ug/kg in humans, which is a clinically relevant dose."
Comment #3: I am wondering how the authors categorized PTTG1 and PLEKHO1 as weak neoantigens. Additionally, based on available literature, PALB2, PTPRS, and ZDHHC16, TP53 have shown potential in eliciting immune responses in breast cancer. I am curious whether the authors observed any relevant expression of these antigens following rhIL-7-hyFc treatment, either alone or in combination with DNA-PCV.
Reply: Cancer neoantigens are unique to each individual tumor. For the studies presented here, we identified cancer neoantigens in the E0771 model using tumor/normal exome sequencing and the pVACtools suite of computational tools.
We have also identified PALB2, PTPRS, ZDHHC16, and TP53 as breast cancer neoantigens (ref 7). However, there is no evidence that these same mutations exist in the E0771 model system. The paucity of shared cancer neoantigens confirms the importance of a personalized approach.
The Reviewer raises an important point about the definition of "weak" neoantigens. There is no clear definition of "weak" and "strong" neoantigens in the literature. We have revised the manuscript to use the terms "relatively weak," and "relatively strong." We describe PTTG1 and PLEKHO1 as "relatively weak" neoantigens, and Lrrc1 as a "relatively strong" neoantigen based on the ELISpot data.
Comment #4: I appreciate the authors' cell-killing experiment using CFSE. However, I am still curious about the underlying mechanism does the cell killing occur through a direct effect, or are immune cells becoming cytotoxic and responding more rapidly to eliminate cancer cells? Additionally, it is not clear which specific cell populations were targeted by the treatments, either with rhIL-7-hyFc alone or in combination with DNA-PCV.
Reply: The in vivo CFSE-based cytotoxicity assay has been described as a functional measure of CD8+ T cell-mediated killing.
In the experiment described in the manuscript, we loaded CFSE-labeled target cells with the minimal MHC class I-restricted peptides corresponding to the PTTG1, PLEKHO1 and Lrrc1 neoantigens.
The enhanced cytotoxicity after DNA PCV + rhIL-7-hyFc is therefore most reasonably explained by CD8+ T-cell mediated killing.
Other immune subsets, such as NK cells or CD4+ cells are not expected to contribute in this context.
As seen in Figure 4b and 4c, rhIL-7-hyFc and DNA PCV alone induce minimal killing of the CFSE-labeled cells.
The combination of DNA PCV + rhIL-7-hyFc results in a significant increase in cytotoxic CD8+ T cell activity, suggesting a direct effect related to expansion of CD8 T cells.
We have clarified this point in Results section 3.3: "Cells labeled with the lower concentration of CFSE were pulsed the minimal MHC class I-restricted peptides corresponding to the PTTG1, PLEKHO1 and Lrrc1 neoantigens. Cells labeled with the higher concentration of CFSE were not pulsed with peptide."
Comment #6: I am unclear about the rationale for using lower concentration CFSE-labeled cells pulsed with E0771, while the higher concentration was not. I suggest the authors clarify this point and provide supporting data to help readers better understand the purpose of this approach.
Reply: In CFSE experiments, two target pools are labeled with CFSE – one target pool with a high concentration of CFSE and one target pool at a low concentration of CFSE. This allows two distinct populations (peptide-pulsed vs. unpulsed) that can be clearly identified by flow cytometry.
We chose to pulse the CFSE lo population with neoantigen peptides as the literature suggests that high concentrations CFSE may be associated with minimal intrinsic cytotoxicity.
To avoid this potential confounder, we chose to pulse the CFSE lo population with neoantigen peptides. The concentrations used for the CFSE lo and CFSE hi populations are well within the typical range observed in the literature.
To address the Reviewer’s concern, we have revised the manuscript to include an additional citation in Methods section [29] and explained the rationale to pulse the CFSE lo population with neoantigen peptides: "In vivo cytotoxicity assays were performed as previously described, and CFSE concentrations are well within the typical range reported in the literature [28,29]."
Comment #7: Although I am partially convinced by the data showing an increase in Lrrc27 neoantigen-specific CD8⁺ T cells with PCV alone (2.5%), and a further five-fold increase when combined with rhIL-7-hyFc, I would like to see whether the authors also evaluated rhIL-7-hyFc alone. In addition, it would be valuable if the authors could provide this dataset in a vaccination setting (Section 3.3).
Reply: We did evaluate the impact of rhIL-7-hyFc alone. This data is actually included in Figure 3b, although the rhIL-7-hyFc group data points are partially hidden by the X-axis. rhIL-7-hyFc alone contributed very little to neoantigen-specific killing. Because of this, it may be difficult to distinguish between the X-axis and the rhIL-7-hyFc data points. We have revised the Figure Legend to bring this to the attention of the reader.
Comment #8: In section 3.4, the authors show an increase in Lrrc27 neoantigen-specific CD8+ T cells. I am curious whether the authors also examined other immune cell types, such as macrophages or regulatory T cells, in this context. Could the authors provide information on the status of other Lrrc27-specific immune populations? Additionally, given the promising increase in Lrrc27-specific CD8+ T cells, it would be helpful to know whether these cells exhibit enhanced cytotoxic activity compared to non-specific CD8+ T cells.
Reply: In this study, our efforts focused primarily on CD8 T-cells because they are the most important MHC-class I-restricted effector cells and can be effectively measured through ELISpot and CFSE cytotoxicity assays.
In addition, IL-7 and rhIL-7-hyFc are known to have an important impact on CD8 T cell-mediated immune responses [21].
We did not conduct additional analysis on macrophages or regulatory T-cells in this context, and we acknowledge that this is an important area for future investigation. Thus, we have revised the Discussion to clarify this limitation: "Lastly, an important area for future study will include a broader investigation of other immune subsets to determine whether rhIL-7-hyFc modulates PVC stimulated immune populations beyond CD8 T-cells."
Comment #9: Some of the figures appear blurry and unclear; improving their clarity would enhance data presentation. Additionally, the discussion section is relatively weak and would benefit from a more thorough justification of the conclusions based on the presented evidence. Finally, there are some grammatical errors and redundancies throughout the manuscript that should be carefully revised.
Reply: The Figures provided to Cancers are of high quality. We suspect that they appear blurry because they were imported into a Word document to expedite review. We will work with the Editorial office to ensure that the final manuscript contains the original high-quality Figures.
In terms of the grammatical errors, we are submitting the manuscript to a professional writing service at Washington University in St. Louis. This service professionally edits manuscripts to ensure the most clear and concise English presentation is used.
Comment #10: Since this manuscript focuses on neoantigen-specific vaccine development, particularly in breast cancer, I suggest that the authors cite the following relevant article, which specifically addresses neoantigens and vaccine strategies in TNBC. Mukherjee, S., & Chatterjee, K. (2024). DNA damage response and neoantigens: A favorable target for triple-negative breast cancer immunotherapy and vaccine development. In International Review of Cell and Molecular Biology (Vol. 389, pp. 104-152). Academic Press. https://doi.org/10.1016/bs.ircmb.2024.05.001.
Reply: We have incorporated the suggested reference in our manuscript, reference #6.
Reviewer 2 Report
Comments and Suggestions for Authors
This manuscript presents a preclinical study investigating the adjuvant potential of long-acting recombinant IL-7 (rhIL-7-hyFc) in combination with personalized cancer vaccines (PCVs) for enhancing anti-tumor immune responses. The study utilizes a murine breast cancer model (E0771) to evaluate the impact of this combination on neoantigen-specific T cell immunity, memory responses, tumor protection, and tumor-infiltrating lymphocytes (TILs).
Critique: The manuscript has substantial strengths and significance. The manuscript is overall well-organized, and the figures are clear and support the presented results effectively. The study addresses the need for stronger and more durable immune responses, especially to weaker neoantigens. The study utilizes a murine breast cancer model (E0771) to evaluate the impact of this combination on neoantigen-specific T cell immunity, memory responses, tumor protection, and tumor-infiltrating lymphocytes (TILs). The findings may have clinical applications for lymphopenic cancer patients. Overall, this manuscript presents compelling preclinical evidence for the synergistic potential of rhIL-7-hyFc and personalized cancer vaccines. While some limitations exist, particularly concerning the single-dose administration and the scope of the animal model, the study provides an innovative immunization approach that may have impact in the clinical setting.
Comments:
- The study is conducted entirely in a single murine breast cancer model (E0771) in C57BL/6 mice. While providing proof-of-concept, the findings may not be generalizable to other cancer types or genetic backgrounds, or to human patients. Different tumor microenvironments and immunogenicity profiles could influence the observed effects. Relevant discussion should be included. At minimum this needs to be discussed.
- rhIL-7-hyFc was administered only once. This may be a limitation, as clinical trials typically involve multiple doses. It is not clear how this technology could be translated into the clinical setting.
- The specific characteristics or ranking of immunogenicity for Pttg1 and Plekho1 compared to Lrrc27 could be further examined. More quantitative data or a clearer definition of strong versus weak neoantigens is needed.
- The results primarily focus on CD8+ T cell responses and cytotoxicity. While CD8+ T cells are crucial for direct tumor killing, the role of CD4+ helper T cells in supporting durable CD8+ responses and memory is significant. Including data on CD4+ T cell responses would provide a more complete picture of the immune activation.
- In Figure 4, the rhIL-7-hyFc treatment was given during the T cell expansion phase (Day 4) due to tumor growth kinetics, differing from the optimized timing (Day 13, contraction phase) for durable memory responses. It is not clear whether this early administration would also lead to prolonged responses and tumor control in a therapeutic setting, which was the main finding in prophylactic settings. Analysis of cytokine profiles in the TME could enhance the study.
- The manuscript does not present long-term survival data in a therapeutic model. This would be a crucial endpoint for potential clinical translation. Is there any information on this?
- The discussion mentions lymphoid expansion evidenced by an increase in spleen size among combination rhIL-7-hyFc PCV treated groups (data not shown). Providing supplementary data for this observation would strengthen the claim.
Author Response
Reviewer #2 Feedback:
Comment #1: The study is conducted entirely in a single murine breast cancer model (E0771) in C57BL/6 mice. While providing proof-of-concept, the findings may not be generalizable to other cancer types or genetic backgrounds, or to human patients. Different tumor microenvironments and immunogenicity profiles could influence the observed effects. Relevant discussion should be included. At minimum this needs to be discussed.
Reply: We agree with Reviewer #2 that our findings may not be generalizable to other cancer types/models/patients. We have revised the title of the manuscript as noted above in response to Reviewer #1.
We have also expanded the paragraph in the Discussion discussing limitations to reflect this limitation.
Specifically, we added "Our findings provide proof-of-concept that IL-7 enhances the impact of PCV. However, our data were obtained in a single mouse breast cancer model and may not be generalizable to other models or to patients."
Comment #2: rhIL-7-hyFc was administered only once. This may be a limitation, as clinical trials typically involve multiple doses. It is not clear how this technology could be translated into the clinical setting.
Reply: Reviewer #2 is correct that rhIL-7-hyFc was given only once in our animal study. As we clarified in the Discussion, the main reason for the single administration was to avoid a neutralizing response to the rhIL7-hyFc compound. However, this is not a concern when the drug is administered to patients; in fact, rhIL-7-hyFc is typically administered to patients at multiple timepoints during a treatment cycle, often at 6-12 week intervals [24,42,43].
Comment #3: The specific characteristics or ranking of immunogenicity for Pttg1 and Plekho1 compared to Lrrc27 could be further examined. More quantitative data or a clearer definition of strong versus weak neoantigens is needed.
Reply: We have discussed this issue in response to Reviewer #1 above.
Of note, our neoantigen identification algorithms does not suggest clear differences between the three antigens. However, T cell responses to these antigens clearly differ. Given that there are no clear definitions in the literature related to "strong" and "weak" neoantigens, we have revised the manuscript and now use the terms "relatively strong" and "relatively weak."
Comment #4: The results primarily focus on CD8+ T cell responses and cytotoxicity. While CD8+ T cells are crucial for direct tumor killing, the role of CD4+ helper T cells in supporting durable CD8+ responses and memory is significant. Including data on CD4+ T cell responses would provide a more complete picture of the immune activation.
Reply: We agree with this point by Reviewer #2. In our clinical trials (ref 7) and unpublished observations), we have observed robust CD4 T cell responses to neoantigens and we intentionally include candidate neoantigens into PCVs based on their predicted CD4 T cell immunogenicity.
However, all of the immune responses in the E0771 model are all CD8 T cell responses.
We tested the top 13 neoantigen candidates for immunogenicity in an unbiased manner, i.e. using synthetic long peptides that can induce either CD4 or CD8 immune responses.
In ELISpot assays, only three neoantigens were capable of inducing an immune response (Lrrc27, Pttg1, and Plekho1). In all three cases, the immune response was a CD8 T cell response.
We have expanded the Discussion by adding the following sentences to the paragraph on limitations: "The immunogenic neoantigens in the E0771 model all elicited CD8 T cell responses. Although rhIL-7-hyFc is also known to enhance CD4 T cell responses, we were not able to test this in the E0771 model system."
Comment #5: In Figure 4, the rhIL-7-hyFc treatment was given during the T cell expansion phase (Day 4) due to tumor growth kinetics, differing from the optimized timing (Day 13, contraction phase) for durable memory responses. It is not clear whether this early administration would also lead to prolonged responses and tumor control in a therapeutic setting, which was the main finding in prophylactic settings. Analysis of cytokine profiles in the TME could enhance the study.
Reply: Our studies suggest rhIL-7-hyFc enhances PCV responses when given at day 4 or at day 13. However, due to rapid tumor growth kinetics, administration of rhIL-7-hyFc in the treatment setting had little to no impact.
We did not perform cytokine analysis in the TME.
Comment #6: The manuscript does not present long-term survival data in a therapeutic model. This would be a crucial endpoint for potential clinical translation. Is there any information on this?
Reply: As mentioned above, rapid tumor growth kinetics in the E0771 model prevented meaningful studies of PCV + rhIL-7-hyFc treatment in the therapeutic setting. Nonetheless, our studies provide rationale for the treatment of patients with no evidence of disease at high risk of disease recurrence, a common clinical scenario currently being explored in early phase PCV studies.
Comment #7: The discussion mentions lymphoid expansion evidenced by an increase in spleen size among combination rhIL-7-hyFc PCV treated groups (data not shown). Providing supplementary data for this observation would strengthen the claim.
Reply: We added a new supplementary figure (S2) that illustrates the increased spleen size in mice receiving rhIL-7-hyFc (with or without PCV).